



# Drought lacunarity around the world and its classification

Robert Monjo[1,2], Dominic Royé[3,4], and Javier Martin-Vide[5]

[1]Climate Research Foundation (FIC), C/Gran Vía 22 (duplicado), 7, 28013 Madrid, Spain
[2]Department of Algebra, Geometry and Topology, Complutense University of Madrid, Plaza Ciencias, 3, 28040 Madrid, Spain
[3]Department of Geography, University of Porto, Via Panorâmica, 4150-564 Porto, Portugal
[4]Department of Geography, University of Santiago de Compostela, Praza da Universidade 1, 15782 Santiago de Compostela, Spain
[5]Climatology Group, Department of Physical Geography, University of Barcelona, C/Montalegre 6, 08001 Barcelona, Spain

**Correspondence:** Robert Monjo (robert@ficlima.org)

**Abstract.** Drought duration strongly depends on the definition thereof. In meteorology, dryness is habitually measured by means of fixed thresholds (e.g. 0.1 or 1 mm usually define dry spells) or climatic mean values (as is the case of the Standardised Precipitation Index), but this also depends on the aggregation time interval considered. However, robust measurements of drought duration are required for analysing the statistical significance of possible changes. Herein we have climatically classi-

fied the drought duration around the world according to their similarity to the voids of the Cantor set. Dryness time structure can be concisely measured by the n-index (from the regular/irregular alternation of dry/wet spells), which is closely related to the Gini index and to a Cantor-based exponent. This enables the world's climates to be classified into six large types based upon a new measure of drought duration. We performed the dry-spell analysis using the full global gridded daily Multi-Source Weighted-Ensemble Precipitation (MSWEP) dataset. The MSWEP combines gauge-, satellite-, and reanalysis-based data to

provide reliable precipitation estimates. The study period comprises the years 1979-2016 (total of 45165 days), and a spatial resolution of 0.5°, with a total of 259,197 grid points. Data set is publicly available at https://doi.org/10.5281/zenodo.3247041 (Monjo et al., 2019).

## 1 Introduction

Drought depends mainly upon the sector affected and the timescale considered (Wilhite and Glantz, 1985; Crausbay et al., 2017). Focusing on the timescale, we usually distinguish between dry spells (daily timescale) and negative anomalies, commonly represented by monthly or yearly indexes such as the Standardised Precipitation Index, the Standardised Evapotranspiration Index or the Palmer Drought Severity Index among others (Vicente-Serrano et al., 2010, 2015). Alternation between dry and wet events presents self-similarity (characteristic of fractal objects), in the same manner that the Cantor set alternates

points with gaps; this is also known as *lacunarity* (Martínez et al., 2007; Feng et al., 2015; Dayeen and Hassan, 2016; Lucena et al., 2018). According to Mandelbrot, fractality can be found by measuring. He noted that, the more accurate the measurement





ruler, the more infinite the British coastline appears to be, since the immeasurable curves of the coast situate it betweena line (one dimension) and a surface (two dimensions), i.e. with a fractal or fractional dimension (Mandelbrot, 1967).

A commonly used method for measuring the dimension of fractal objects involves *box counting*, which is similar to using a ruler for measuring a coastline. Given an object embedded in an $N$-volume ($N$=1, length; $N$=2, area; $N$=3, volume; etc.), the method consists of covering the object several times, using unitary ($N$–1)-volume boxes of different sizes $r$ for each completed covering, and counting how many covering boxes are required in each case (Olsson et al., 1992; Sakhr and Nieminen, 2018). As the box size becomes smaller, the total ($N$–1)-volume of the fractal object tends towards the infinite rather than converging towards a finer value, and the $N$-volume is zero. For instance, the Cantor set is embedded in the [0, 1] segment with infinite 30 (0-dimensional) points, but its total (1-dimensional) length is zero. Formally, an object (embedded in an $N$-volume) possesses a fractal (non-integer) dimension $B$ between $N$–1 and $N$ if there exists a well-defined $B$-volume $V = M_r r^B$, where $M_r$ is the number of boxes with size $r$ (Imre and Bogaert, 2006). A well-defined $B$-volume means that V and B remain constant for small values of $r$.

Another related measure involves the Lyapunov exponent, which indicates the rate of separation of infinitesimally close 35 trajectories, or involves the inverse, sometimes referred to as Lyapunov time, since it indicates the timeexpected to become a chaotic trajectory (Boeing, 2016; Kuznetsov, 2016; Gaspard, 2005; Bezruchko and Smirnov, 2010). The Hurst exponent is also related to the fractal dimension of chaotic time series, providing possible long-term memory throughout autocorrelation (Mandelbrot, 1985; Feder, 1988; Yu et al., 2015).

The fractal behaviour of dry spells can be observed in a Richardson's log-log plot of cumulative dry durations with regard 40 to different unit durations (Sen, 2008; Meseguer et al., 2017). Similarities with the Cantor set (positive Lyapunov exponents) were found for dry-spell sequences in Europe (Lana et al., 2010).

According to a multifractal analysis of the standardised precipitation, power-law decay distribution describes well the probability density function of return intervals of drought events (Hou et al., 2016). The Hurst exponent was also used to analyse the fractal persistence of the Palmer Drought Severe Index, providing values close to 1 (i.e. long-term positive autocorrelation) 45 throughout Turkey (Tatli, 2015). The concept of persistence of dryness is used by some authors as an early indicator of drought, according to the upper-order Markov Chain model (**?**Lana et al., 2018; López de la Franca Arema et al., 2015).

In addition, the fractal density of wet (or dry) spells can be estimated according to the $n$-index (Monjo, 2016). This index measures the persistence of records (or lengths) of a sequence of wet (or dry) spells similar to how regularity is measured in a Lorenz curve, whilst preserving the time structure of the events. A value of $n$ <0.5 implies that a time series is persistent 50 (consecutive similar values), while for n >0.5 the time series is anti-persistent. This regularity measure is closely related to the Shannon entropy, the Gini index ($G$) and the box-counting dimension of rainfall (Monjo, 2016; Monjo and Martin-Vide, 2016). For this reason, the $n$-index constitutes the main measure chosen in our work for analysing drought lacunarity around the world, also compared with a Cantor-based exponent ($C_e$).





## 2 Methods

### 2.1 Dry-spell *n*-index

The main fractal measure was estimated for dry-spell density by means of the *n*-index. For this propose, each spell duration $D_i$ was taken as a precipitation value, considering the minimum value $D_0 = 1$ as the dry value by definition. For instance, let $D = (3, 4, 7, 1, 3, 10, 12)$ be a time series of consecutive dry spells. Subsequently, two independent events ("*spells of spells*") are built around the dry value ($D_0 = 1$) as (3, 4, 7), (3, 10, 12) (Supplementary Fig. A2). In the present study, only dry spells were considered for building events; thus, each separated event is referred to as a "*dry-spell spell*" (DSS).

In a similar way as for precipitation, the maximum accumulated dry-spell duration ($P_i$) of a DSS event is defined as:

$$P_i = \max \left\{ \sum_{j=k}^{k+i-1} D_j \right\}_{k=1}^{N-i+1} \tag{1}$$

where *i* is the number of accumulated events, and *N* is total considered events. For each DSS event, maximum average duration $Y_i$ at *i*-step is:

$$Y_i = \frac{P_i}{i} \tag{2}$$

Therefore, the maximum average duration satisfies a scaling relationship with respect to this event number:

$$\frac{Y_i}{Y_1} = \left( \frac{1}{i} \right)^n \tag{3}$$

where $Y_1$ is the maximum expected dry length per year and *n* is the *n*-index, which is bounded as $d \leq n \leq 1$, i.e. between the fractal dimension (*d*) of the spells considered and the dimension of the time series (Monjo, 2016). The parameters $Y_1$ and *n* were fitted for each DSS and averaged for each timeseries of grid points. Taking Eq. 2 and **??**, maximum accumulated dry-spell duration ($P_i$) is:

$$P_i = Y_1 i^{1-n} \tag{4}$$

Due to the low probability of the longest spells, a high maximum duration $Y_1$ implies a big difference in relation to the previous and subsequent durations, i.e. it implies high values for *n*. Therefore, a statistical link is expected between the probability distributions of $Y_1$ and *n*. In order to test this hypothesis, it suffices to set a distribution for one and a fit for the other. For example, if a distribution over $Y_1$ is set as $1 - 1/Y_1$, we can find a distribution *F(n)* of *n* such as:

$$1 - \frac{1}{Y_1} = F(n) \tag{5}$$

In particular, two-parametric versions of three theoretical distributions were fitted: Exponential ($F1$), Classical Gumbel ($F_2$) and Opposite Gumbel ($F_3$) distributions (Monjo and Martin-Vide, 2016):

$$F_1(n) = 1 - \exp(-\alpha_1 n + \beta_1) \tag{6}$$

$$F_2(n) = \exp(-\exp(-\alpha_2 n + \beta_2)) \tag{7}$$

$$F_3(n) = 1 - \exp(-\exp(\alpha_3 n + \beta_3)) \tag{8}$$





The Akaike Information Criterion was applied to each fitted model using the log-likelihood function according to the equation

$$AIC = -2\log(\hat{L}) + p_m k \tag{9}$$

where $\hat{L}$ is the maximum value of the likelihood function for the model fitted, $p_m$ is the number of parameters in the model, and $k = 2$ is used for the usual AIC, or $k = \log(N)$ (with $N$ equal to the number of observations) for the so-called BIC (Bayesian Information Criterion) (Supplementary Fig. A3).

The results of the dry-spell $n$-index were compared with other dryness fractal measures in each grid point: the Cantor-based exponent ($C_e$), the Hurst exponent ($H$) and the Gini index ($G$), all the cases estimated considering the entire time-series of dry spells $\{D_j\}$. The Gini Index is defined as the area under the Lorenz curve, which describes the relative accumulation of the variable $D_j$ and its cumulative frequency (Monjo and Martin-Vide, 2016). Alternatively, the Hurst exponent measures the possible long memory in time series. Given a set $\{A_i\}$ formed by $T$ anomalies $A_i$ of the variable ($D_j$) and the corresponding set $\{C_i\}$ of $T$ cumulative values $C_i$ from $A_1$ to $A_j$, the Hurst exponent $H$ is obtained from the relation $R/S = k_H \cdot T^H$, where $k_H$ is a constant, $S$ is the standard deviation of the set $\{A_i\}$ and $R$ is the range (i.e. maximum minus minimum value) of the set $\{C_i\}$ (Mandelbrot, 1985; Feder, 1988).

## 2.2 Cantor-based exponent

Finally, the lacunarity of the Cantor set was compared with the frequency distribution of dry-spell durations for a given time-series of $L$ days. To this end, a Cantor-based time series was built using '*segments of zeros*' or gaps $\{G_{kj}\}$ found between consecutive Cantor points (represented by '*segments of ones*') obtained by the $k$-th iteration given by $k = \log(T)/\log(3)$, where $T$ is the length of the binary time series considered. For example, for the first iteration, $k = 1$, only the gap $G_{1i} = T/3$ is obtained; for $k = 2$, three gaps are found, $G_{2i} = \{T/9, T/3, T/9\}$; for $k = 3$, seven gaps $\{G_{3i}\} = \{T/27, T/9, T/27, T/3, T/27, T/9, T/27\}$; and so on. The set of gaps greater than one,

$$\Gamma_k := \operatorname*{sort}_i \left\{ \frac{G_H}{T} : G_H > 1 \right\} \tag{10}$$

was compared with that obtained from the set of dry spells,

$$\Delta := \operatorname*{sort}_j \left\{ \frac{D_j}{L} : D_j > 1 \right\} \tag{11}$$

The value of the iteration k was chosen as the minimum iteration, when the total number of elements of $\Delta$ is less than, or equal to, the total number of elements (cardinal) of $\Gamma_k$, i.e. $|\Delta| \leq |\Gamma_k|$. Finally, we defined a Cantor-based exponent $C_e$ by the quantile-quantile map

$$\Delta = \delta_\epsilon \cdot \Gamma_k{}^{C_e} \tag{12}$$

## 2.3 Data availability

The data used in this study was obtained from the full global gridded daily Multi-Source Weighted-Ensemble Precipitation (MSWEP) dataset (Beck et al., 2017). The MSWEP combines gauge-, satellite-, and reanalysis-based data to provide reliable

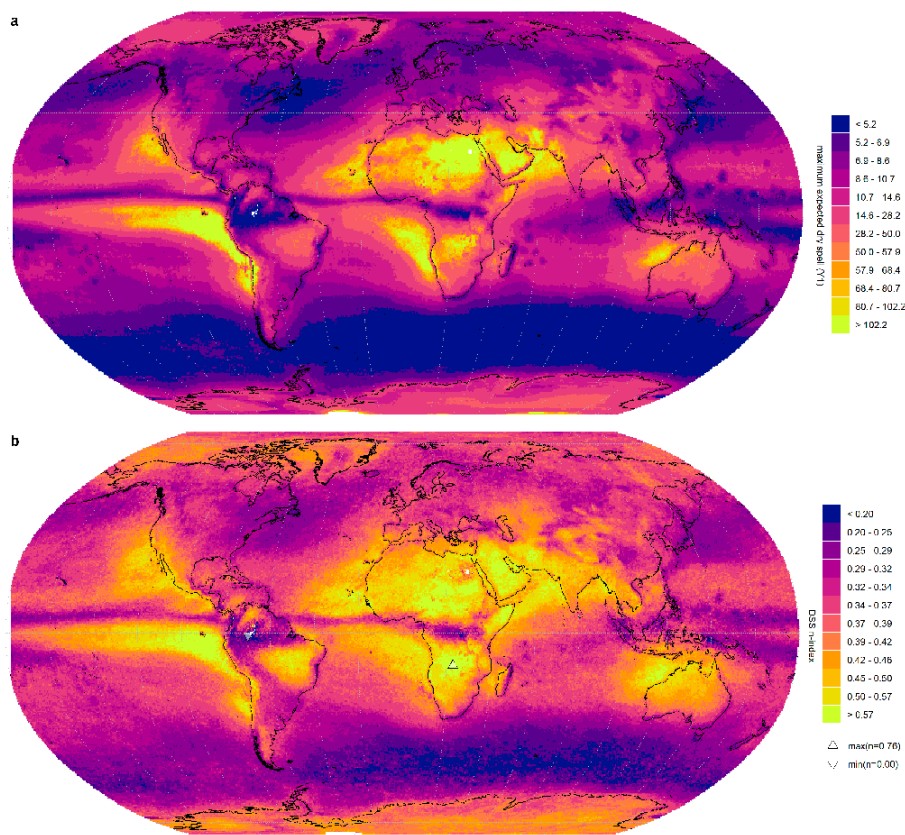

**Figure 1.** Spatial distribution of regularity of the lacunarity, averaging for all DSS events: a) Maximum expected dry spell ($Y_1$), and b) DSS *n*-index

precipitation estimates. The study period comprises the years 1979-2016 (total of 45165 days), and a spatial resolution of 0.5°,
115 with a total of 259,197 grid points (Supplementary Fig. A1).

## 3 Results

### 3.1 Drought patterns detected

By analysing Dry Spell Spells (DSS), the first overview of the spatial distribution of the DSS *n*-index is given by seven quasi-latitudinal bands (Fig. 1): Three low-value bands in the equatorial zone and at medium latitudes in both hemispheres, and
120 four high-value bands in the two tropical areas and in the two polar areas. These bands generally correspond to the large climatic areas of the world. Indeed, there is a statistically significant relationship between annual dry days and the *n*-index (Supplementary Fig. A1).



More specifically, rainforests like those in the Amazon, the Congo or Southeast Asia present values of $n$ <0.3. This is consistent with the low values also found for other rainforest zones such as those in Madagascar, Central America and South America. This is due to the high degree of persistence of very short dry spells, alternating with very frequent wet days. Low values are also provided by the domain regions due to the action of the polar jet streams in both hemispheres. This is the case of the zonal flux in the Southern Ocean and the eastern fringes of North America and Asia.

High values of the index ($n$ >0.4) are presented in the tropical and subtropical regions, due to the effect of the subtropical anticyclones, particularly in the deserts. The Savannah regions show the highest values of n, which involve a long dry spell followed by shorter dry events. The Polar Regions score a secondary maximum of n due to the usually long but irregular dry events. Similar spatial results are found if $G$ and $C_e$ are used (Fig. 2), while the Hurst exponent displays noisier patterns around the world, very close to 0.5, as described by other studies for Europe Martínez et al. (2007); Lana et al. (2010). Most of the cases show values $n$ <0.5; similarly, approximately 80% of the $G$ values are lower than 0.5. Both results indicate the existence of a noteworthy degree of homogeneity in the distribution of the dry spells along the time series. In fact, the Hurst exponent is close to 0.5 (i.e. the associated fractal dimension could be 1.5), which can be interpreted as an equilibrium between positive and negative autocorrelations.

## 3.2 Drought classification

Three main sets can be identified according to the predominance of low (L), medium (M) or high (H) values of the DSS $n$-index: Type S when $n$ <0.3, Type M if $n$ is within the interval (0.3, 0.4), and Type L for $n$ >0.4. For the three main types, it is advisable to distinguish between the alternation with longer ($\ell$) or shorter (s) wet events; for example, considering the threshold of three consecutive wet days (Supplementary Fig. A1). Therefore, six large drought types can be defined based upon the combination of both criteria (Fig. 3):

- Type L$\ell$ . Occurrence of very short dry spells alternating with longer wet spells.

  - Tropical examples: The main rainforest cores of the world (within the Amazon, the Congo and Southeast Asia among others).

  - Subpolar examples: The Southern Ocean and some regions of the North Atlantic and North Pacific oceans.

- Type Ls. Occurrence of very short dry spells alternating with short wet spells. Examples: northeast America and northeast Asia, especially Japan.

- Type M$\ell$ . Median dry spells alternating with longer wet spells. Examples: The Artic Ocean and North Asia.

- Type Ms. Median dry spells alternating with shorter wet spells. Examples: the Central area of North America and the temperate regions of the Atlantic and Pacific Oceans.

- Type H$\ell$ . Occurrence of very long dry spells alternating with long wet events. Examples: The tropical savannah regions of Africa, Mexico, Central Brazil, India (monsoon climate), Southern China and Northern Australia.

**Figure 2.** Spatial distribution of other fractal measures applied to dry spells: a) Hurst exponent, b) Gini index, c) Cantor-based exponent.

– Type Hs. Occurrence of very long dry spells alternating with short wet events. Examples: All the desert regions around
the world, including the eastern fringe of the tropical oceanic areas.

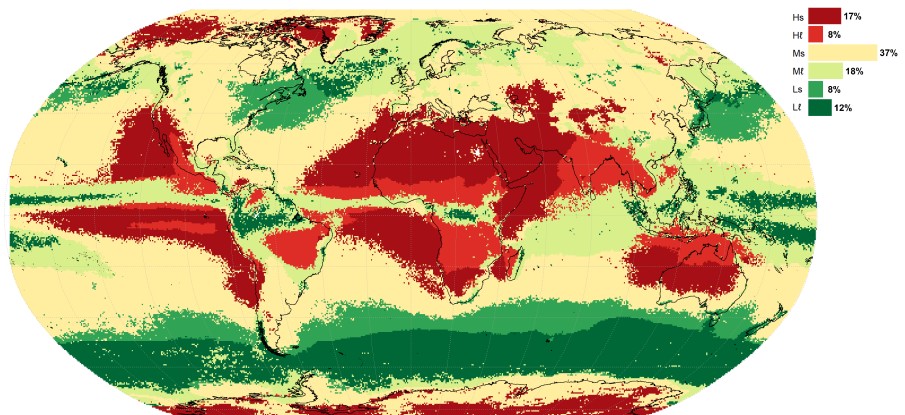

**Figure 3.** Climatic classification of meteorological droughts around the world: Regions with low (L), medium (M) or high (H) values of the DSS $n$-index, alternating with longer ($\ell$) or shorter (s) wet events.

### 3.3 Drought lacunarity

Since the total (1-dimensional) length of the Cantor set is zero, the total length of the complementary gaps is equal to one. That is, following the analogy between the drought duration and the Cantor set lacunarity, the total duration of a dry spell series approaches one when the size of the measurement box is accurate. For instance, one can find dry days in a wet month, and on rainy days, there can be several hours with no rain. If a ground point is used for measurement, the duration of a rain drop hitting the ground (from leading surface to trailing surface) tends towards zero and thus the dry pauses are distributed paradoxically throughout an entire rainy day.

According to this idea, the dimension of the drought duration is practically one (the length of the time series), and the box-counting dimension therefore makes more sense for measuring rainfall duration than for estimating drought duration. However, the lacunarity of the drought can be analysed by means of other measures, such as the Gini index ($G$) and the Cantor-based exponent ($C_e$), both of these related to the frequency of dry spell durations. In particular, the Cantor-based exponent indicates how likely it is to find longer dry spells over time. For instance, if $C_e \sim 0$, all dry spells will present similar lengths and therefore the standard deviation will be constant (i.e. extreme values are normally distributed). However, if $C_e \sim 1$, the distribution of lengths is similar to the Cantor lacunarity, and the standard deviation will therefore increase over time. In this case, extreme dry spells show a linear increase for longer time series (in the same way that the maximum Cantor gap is set at 1/3 of the total length), and the Gini index also tends to approach 1. Indeed, the correlation between the $G$ and $C_e$ is $R^2 = 0.86$ ($p$-value <0.0001) and the approximation $C_e = G$ only provides an error of 10%, thus reinforcing the lacunar interpretation of the dryness.





## 4 Discussion

The DSS $n$-index provides information on the structure of drought lacunarity, in particular measuring the probability of regularity (if $n \sim 0$) or irregularity (if $n \sim 1$). Regular values of dry spells indicate that similar dry-spell lengths are usually consecutive. Accordingly, irregular values imply that long dry spells are followed by much shorter dry sequences. It should be kept in mind a high degree of irregularity is correlated with the longest dry spells (Supplementary Fig. A3).

In short, the time patterns of the dryness of a climate can be characterised in a simple manner by means of the $n$-index. This
indicator represents a synthetic DSS, computed by averaging all the DSS events of a considered time series. In each DSS, a relative distribution can be observed of consecutive dry spells and the maximum dry-spell duration. For example, the highest values of the DSS $n$-index (observed in parts of Africa) imply a high accumulation of dry-spell durations with a low number of events considered; whereas the lowest values of the DSS $n$-index (found in the the North-Western Amazon) point to a constant and linear accumulation of dry spells presenting the same (short) duration (Supplementary Fig. A4).

The DSS $n$-index therefore provides information on the relative distribution of dry spells, on the longest duration and on the total accumulated duration. Indeed, the start/end of a synthetic DSS event can be established by the number '$i$' of dry-spell events, so that the averaged maximum duration Yi is a particular threshold (e.g. one day, which is defined as the dry threshold of a DSS event). This idea enables the concept of a DSS event to be employed as an alternative definition of meteorological drought, with diffuse borders that, paradoxically, are well defined by the $n$-index. For instance, if $Y_i = 0$ is considered as drought
borders, the total duration of the dryness coincides with the total length of a time series (as the length of the supplementary gaps of the Cantor set). In this case, the difference between one drought or another is given by the decay rate of the maximum dry spell (well measured by the $n$-index).

As in (fractal) wet spells, the behaviour of dryness is self-similar on all timescales, that is to say, dry spells can be used at daily, monthly, yearly resolution, etc., considering specific dryness thresholds. This is guaranteed by the goodness-of-fit of
the $n$-index model ($p$-value <0.0001). The proposed mathematical definition is complementary to the previous ideas on the persistence of dryness, for example, according to upper-order Markov chains26. Indeed, the order of chains depends on the alternation and frequency of short and long dry spells, as with the rest of the measures (Gini index, Hurst exponent, $n$-index, etc.).

## 5 Conclusions

As a principal conclusion, the study demonstrates that drought lacunarity can be analysed with the use of self-similarity features obtained from the DSS $n$-index. This measure is useful for characterising the temporal and spatial patterns of drought, which are consistent with the climatic features established worldwide. For instance, localities presenting very frequent wet spells alternating with short dry spells provide low values of the DSS $n$-index (this is the case, for example, of the Amazon, the Congo and other rainforests). This results from an accumulation of isolated short dry spells, all presenting the same duration.
Additionally, places with longer dry spells (such as deserts) scored higher values on the DSS $n$-index. This result is logical because the $n$-index is strongly correlated with the maximum and average lengths of dry spells, as well as with the Gini index.





Consequently, we arrived at a second conclusion: the methodology developed can be used to classify typologies of meteorological drought. Indeed, six climate types have been proposed; these result from the combination of three classes of DSS $n$-index values (Type L when n <0.3, Type H for $n$ >0.4 and Type M otherwise), and two sub classes based on the alternation

with longer ($\ell$) or shorter (s) wet events. This classification could be useful with regard to monitoring possible changes in drought patterns around the world within the context of climate change (Monjo et al., 2016; Craine et al., 2013; Madakumbura et al., 2019).

The third and most important outcome refers to the fact that consideration of dryness lacunarity provides a better understanding of drought duration and helps to predict when droughts start and finish. In particular, the value of the DSS $n$-index is

related to the effective duration (according to borders determined by a threshold of the "*minimum expected dry spell*"). Indeed, the best analogy with regard to understanding the features of the DSS is Cantor lacunarity, i.e. total dry spells are almost equal to the length of the entire series (in the same manner that the total length of Cantor voids is equal to one).

## 6 Code and data availability

The datasets from Figures (1-3, A1) can be accessed through: Monjo et al. (2019). Drought lacunarity around the world and its

classification (Version v0.1) [Data set]. doi: https://doi.org/10.5281/zenodo.3247041. The used software code (R programming language) with a point example can be accessed through: Monjo et al. 2019 at github https://github.com/robertmonjo/drought.

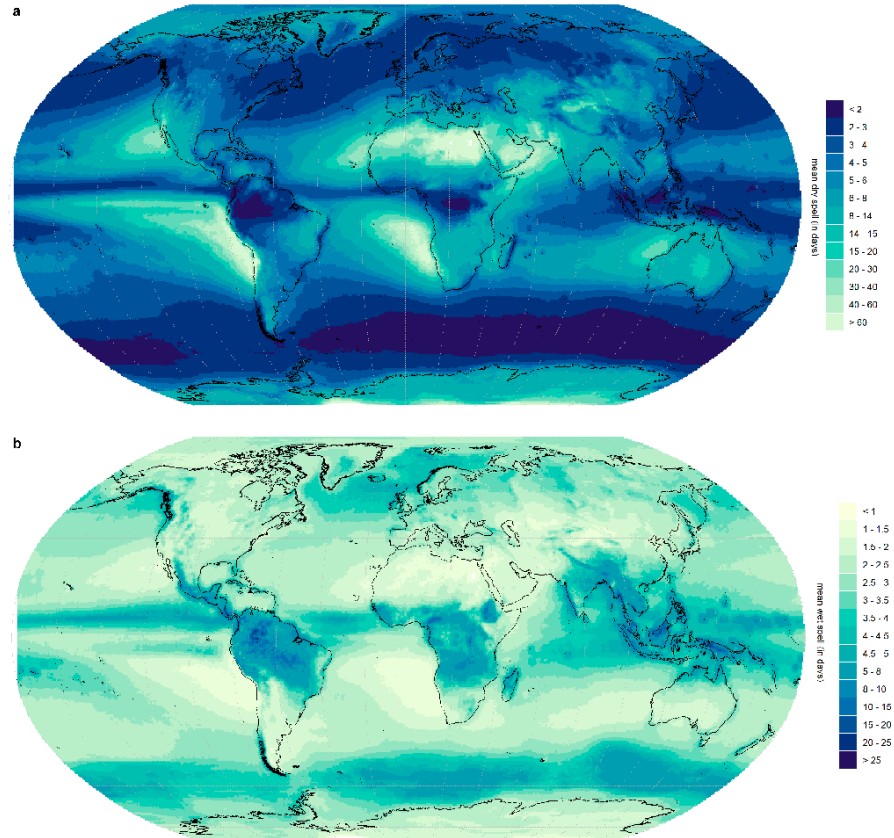

**Figure A1.** Spatial distribution of mean dry spell (a) and mean wet spell (b) at a spatial resolution of 0.5°×0.5°according to the MSWEP dataset (1979-2016).

**Appendix A**

Note that the mean dry spell (MDS) and the mean wet spell (MWS) are geographically distributed with a strong correspondence between the high (low) values of the one and the low (high) values of the other. A general zonal pattern is observed, albeit with a clear dissymmetry between the western facades of the continents and the eastern oceans, in tropical-subtropical latitudes, with high values of the MDS (low values of the MWS), and their responses, with low (high) values. The highest values of the MDS are concentrated in the tropical-subtropical zones, especially in the Sahara desert and its continuation in the Arabian desert, the Persian Gulf and the waters of the Indian waters of the Indian Ocean, in lower California, in northern Chile, on the coast of Peru and its waters, in the Atlantic Ocean off the coast of Namibia and towards the west-northwest of Australia (bordering with Antarctica). The most similar values between both indices, specifically with high values in both, are observed in the Asia-Pacific region (monzonic region), reflecting the division of the year into two halves, the rainy summer, with long rains, and the dry winter, with an absence of precipitation for many days in a row. This pattern can be detected in other areas



presenting a humid-dry tropical climate, such as the southernmost area of the Sahel, north-eastern Brazil, an area in Africa (south of the equator) and northern Australia. The SW-NE diagonals of mid-latitudes with decreasing values of the MDS in the

Atlantic and the Pacific are clearly appreciated, according to the westerlies and the track of the low pressure systems.



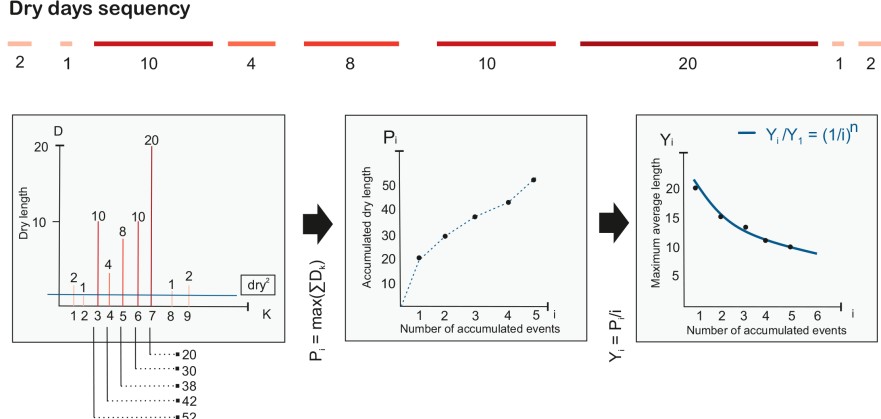

**Figure A2.** Method for defining DSS events and their corresponding n-index.

For instance, let $D = (2, 1, 10, 4, 8, 10, 20, 1, 2)$ be a time series of consecutive dry spells. A DSS event is built around the minimum value ($D_0 = 1$) as $(10, 4, 8, 10, 20)$. Subsequently, maximum accumulated dry durations are $P_i = (20, 30, 38, 42, 52)$, and the maximum averaged dry durations are $Y_i = (20, 15, 12.7, 10.5, 10.4)$, which can be fitted over the number of events ($i$) according to Eq. 3.



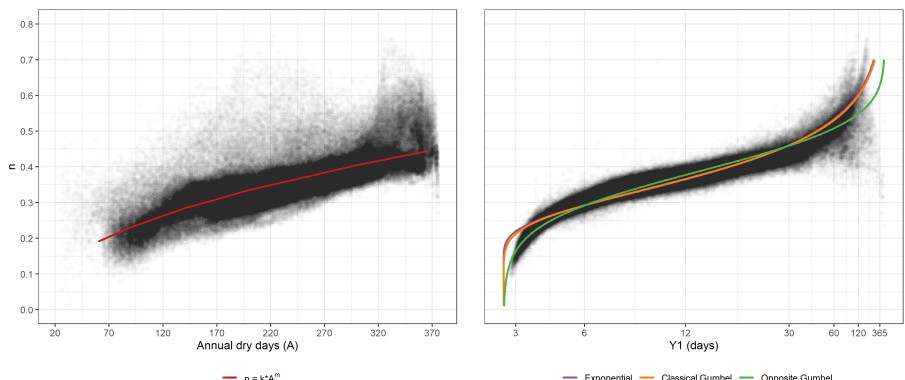

**Figure A3.** Statistical relationship between: a) Annual dry days ($A$) and the DSS $n$-index; b) maximum expected dry spell ($Y_1$) and the DSS $n$-index.

The fitted parameters for the first curve are $k = 0.02811(3)$ and $m = 0.4677(8)$ ($R^2 = 0.67$, $p$-value <0.0001). In the second panel, the fitted curves are Exponential ($\alpha_1 = 0.997(3)$, $\beta_1 = –9.543(7)$, AIC $\approx$ BIC = $˘1.4 \cdot 10^6$, $R^2 = 0.91$, $p$-value <0.0001), Gumbel ($\alpha_2 = 1.225(3)$, $\beta_2 = –10.01(1)$, AIC $\approx$ BIC = $˘1.2 \cdot 10^6$, $R^2 = 0.88$, $p$-value <0.0001) and Opposite Gumbel ($\alpha_3 = 3.829(2)$ and $\beta_3 = –0.5347(9)$, AIC = $˘1.1 \cdot 10^6$, $R^2 = 0.87$, $p$-value <0.0001), according to the Eqs. 6-8.



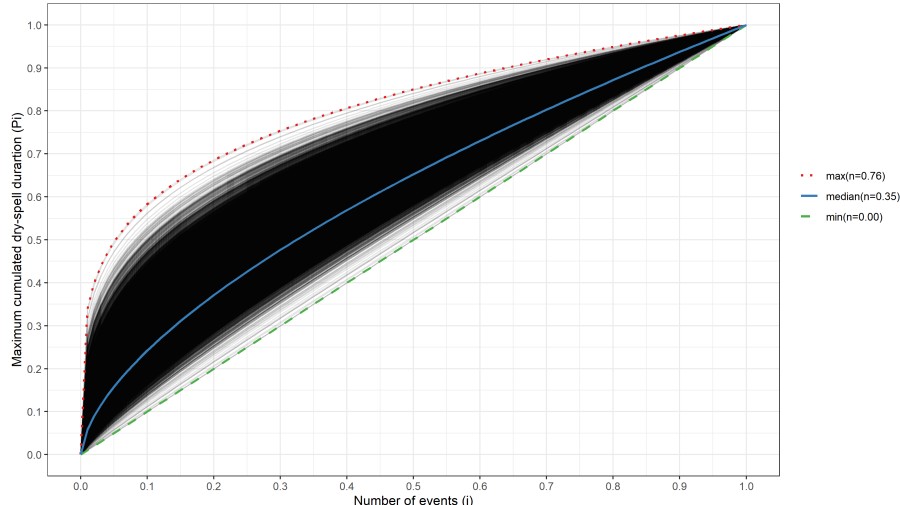

**Figure A4.** Fitted curves of DSS events (Eq. 4), according to the maximum accumulated dry-spell duration ($P_i$) over the number of events considered ($i$), both normalized according to the total values.

Note that the minimum DSS $n$-index corresponds to the Amazon, where almost every DSS event lasts for only one day, and
the accumulation of dry spells ($Y_i$) is therefore very regular. Furthermore, the maximum $n$-index (close to the Zambezi River) represents an average DSS event with a maximum dry spell of 33% of the total accumulated dry-spell length (most of the entire time series), and successive dry spells with shorter durations (as in the Cantor gaps).



*Author contributions.* The Cantor-based exponent and the $n$-index were designed by R.M, all the calculus and figures were created by D.R., while R.M and J.M.-V. interpreted the results and R.M. wrote the manuscript. All authors read and approved the final manuscript.

*Competing interests.* The authors declare that they have no conflict of interest.

*Acknowledgements.* We are grateful for the support provided by the RESCCUE project, which received funding from the European Research Council under the European Union's Horizon 2020 research and innovation programme (grant agreement no. 700174). We also wish to acknowledge the support received from the Spanish projects CGL2017-83866-C3-2-R and Climatology Group 2017 SGR 1362. We ap-
preciate the interest in our research shown by the Water Research Institute of the University of Barcelona and by the Department of Algebra, Geometry and Topology of the Complutense University of Madrid.



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
