# Peer review of "Drought lacunarity around the world and its classification"

_Earth System Science Data, 2019_

## Referee Comment (RC1) · Hasan Tatli (Referee) · 2 Sep 2019

Journal: Earth System Science Data

Manuscript Number: essd-2019-115

Title: Drought lacunarity around the world and its classification distribution patterns, trends, and return level estimates (by Monjo et al.)

Article Type: Preprint (Discussion)

Keywords: There is no keyword

In the manuscript, authors have climatically classified the drought 5 duration around the world according to their similarity to the voids of the Cantor set. Dryness time struc-

ture can be concisely measured by the n-index (from the regular/irregular alternation of dry/wet spells), which is closely related to the Gini index and to a Cantor-based exponent. The paper is interesting. To my thought it can be published after some "major revision", corrections and contributions as given in the following.

Absract:

The first sentence can be written more literally because it does not feel good. In the "Abstract", the authors were supposed to give a few main conclusions. 1. Introduction The lacunarity pertains to both gaps and heterogeneity of the attractor the related dynamical system. The term of "lacunarity" should be explained in the "Introduction" section with examples from the literature. It also needs to be explained how it is associated with drought. Because drought is not only related to a dynamic system (time interval or spell), but to the interaction of many meteorological variables

2. Methods

This section with its subsections should be rewritten in a way that is understandable to an average reader.

1. Results

This section can be written much more appropriate. Maybe a few minor corrections can be added. For example, hydro-climatological interpretations were expected instead of statistical interpretations. This is not a statistical study. In that case, it would be much more meaningful if the hydrological point of view came to the forefront. In addition, the scope of this journal allows the subject of this work.
* * *

---

## Author Comment (AC1) · 17 Sep 2019

Dear Hasan, Thank you very much for the comments. They have been very useful to improve the paper. You can find the changes below.

Abstract ÂńThe first sentence can be written more literally because it does not feel goodÂż This sentence have been changed to: (Changes in manuscript: Page 1, line 1): "The measure of drought duration strongly depends on the definition considered" ÂńIn the "Abstract", the authors were supposed to give a few main conclusionsÂż Additionally, a conclusion have been added to the Abstract. (Changes in manuscript: Page 1, line 8): "To conclude, outcomes provide ability to determine when droughts start and finish"

[Figure]

1. Introduction. ÂńThe lacunarity pertains to both gaps and heterogeneity of the attractor the related dynamical system. The term of "lacunarity" should be explained in the "Introduction" section with examples from the literature. It also needs to be explained how it is associated with drought. Because drought is not only related to a dynamic system (time interval or spell), but to the interaction of many meteorological variablesÂż In our context, drought is used to refer large patterns of dryness in rainfall sequences. To improve the explanation of the ÂńlacunarityÂż concept, we have introduced some sentences : (Changes in manuscript: Page 1, line 20) [...] in the same manner that the Cantor set alternates points with gaps; this is also known as lacunarity. (Changes in manuscript: Page 2, line 43) [...] ÂńIn this sense, dry pauses of the rainfall can be compared with the gap distribution of fractal objects, which is also known as \textit{lacunarity} \citep{Martinez2007,Lucena2018}. The lacunarity analysis is used to characterise 'spatial' patterns (such as invariance, density and heterogeneity) of fractal objects, which represent \textit{attractor} solutions of nonlinear dynamical systems \citep{Plotnick1996,Wilkinson2019}. If a time series of precipitation is solution of the climatic system in a given point, the dryness distribution informs about (climate) average features of the system (e.g. surface convergences/divergences of moisture flows and latent energy fluxes or speed of the hydrological cycle).Âż Plotnick R.E, Gardner R.H., Hargrove W., Prestegaard K., Perlmutter M.A. (1996). Lacunarity analysis: A general technique for the analysis of spatial patterns. Physical review E, Statistical physics, plasmas, fluids, and related interdisciplinary topics 53(5):5461-5468. DOI: 10.1103/PhysRevE.53.5461 Wilkinson M., Pradas M., Huber G., Pumir A. (2019): Lacunarity exponents. Journal of Physics A Mathematical and Theoretical 52(11). DOI: 10.1088/1751-8121/ab0349

2. Methods ÂńThis section with its subsections should be rewritten in a way that is understandable to an average readerÂż Thank you very much for the suggestion. We have rewritten the subsections to reinforce conceptual pathway. The changes are: (page 3, line 64): "The main fractal measure was estimated for dry-spell density by means of the n-index. In a similar way as n-index describes the decrease rate (power

law) of the maximum average intensities of rainfall over time (within a particular me-
teorological event), it also can be applied to analyse how dry-spell lengths decrease
around a maximum value. For this propose, each spell duration... [...]" In a similar
way as for precipitation, Following the same method as in precipitation, the maximum
accumulated (page 4, line 83): "Statistical analysis" (page 5, line 122): "Data availabil-
ity considered" [Note: To distinguish with the section "6. Code and data availability"]
Additionally, lines 89-96 where moved to the end of the methodology section.

1. Results ÂńThis section can be written much more appropriate. Maybe a few minor
corrections can be added. For example, hydro-climatological interpretations were
expected instead of statistical interpretations. This is not a statistical study. In that
case, it would be much more meaningful if the hydrological point of view came to
the forefront. In addition, the scope of this journal allows the subject of this work.Âż
Thank you very much for the suggestions. We would like to emphasize that the paper
has a clear methodological and non-hydrological orientation. The key of the article
is the consideration and utility of the Cantor set, combined of other commonly used
methods, for the objective statistical analysis of the dry/wet sequences. We think
that considering the Cantor set as a model of the temporary behaviour of droughts
is a new conceptual contribution to the analysis of consecutive dry day sequences.
Remember that the Cantor set has a null measure and, on the other hand, is not
empty, or even not countable. This reinforces the fact that although the number of rainy
days can be very large in some regions, always its temporary representation against
dry days is much lower in any climate. In future works we can relate the temporary
behaviour of droughts with the atmospheric causes that produce them. In anticipation,
we have added a brief sentence about a (general) meteorological interpretation of the
results: (pag. 8, line 167): "This classification is consistent with the main atmospheric
circulation belts. For instance, L$\ell$ is strongly linked to the extreme air flow regimes
(inland areas of the equatorial calms and most intense areas of the polar jet streams),
which involve most frequent rainy days due to continuous moisture convergence (see,
e.g., spatial distribution of 10-meter wind regimes in \citet{Possner2017})." Possner,

[Figure]

A., Caldeira, K. (2017): Geophysical potential for wind energy over the open oceans. In: Proceedings of the National Academy of Sciences of the United States of America, 114: 11338-11343.

Please also note the supplement to this comment:
https://www.earth-syst-sci-data-discuss.net/essd-2019-115/essd-2019-115-AC1-supplement.pdf

**Supplement:**

[revised manuscript text omitted]

---

## Referee Comment (RC2) · Serguei G. Dobrovolski (Referee) · 28 Jan 2020

Review of the paper "Drought lacunarity around the world and its classification" by Robert Monjo et al.

The authors introduce new methods for assessing different aspects of meteorological droughts through calculation of specifically constructed indexes. The basis of the work is the set of daily values of precipitation (MSWEP) calculated for 0.5x0.5 deg. grid (1979-2016).

The data used and the methods are, to a large extent, new. In principle, there is a potential of the set of indexes, calculated by the others as for using it in the future. Methods and materials are described sufficiently in the paper.

[Figure]

The data set is accessible via the address given by the authors: https://zenodo.org/record/3247041#.XjBpYxvWj84. However, it seems to me that possible errors within the initial precipitation data set MSWEP are not sufficiently discussed. It is well known that in many regions of the world meteorological stations are sparsely settled, and estimations using satellite data and reanalysis are characterized by considerable errors. At the same time, the authors build their sophisticated mathematical constructions on the basis of theses vague data. The dependence of the results of the work on possible errors in the initial MSWEP data set might be recommended (perhaps, for the future papers of the authors).

One more point. It is important to stress, already in the title of the article that the authors deal only with METEOROLOGICAL droughts. This kind of drought do not necessarily create hydrological drought, and the late is not always related to the agricultural drought, which gives most financial and humanitarian damages. So, in the future it would be interesting to construct indexes, which could be related to the hydrological and agricultural droughts, involving data sets on river discharges and drought damages.

Evidently, the paper in its present state is the subject of discussions as well as all papers describing new ideas. Nevertheless, this work deserves publication. Perhaps, other researchers will find it appropriate to use the paper and data sets in their work, because the data set is usable in its current format and size.

The length of the article is appropriate, it is well structured and clear. The language is consistent and precise. Formulas are correctly defined and used. The figures are of sufficient quality.

Ratings.

Significance: 4 Data quality: 4 Presentation quality: 3.5

---

## Author Response (AR1)

Dear Serguei Dobrovolski,

Thank you very much for your comments. The suggestions have been very useful to improve the applicable orientation of our work. A brief explanation on the reliability of the data has been added (dry spell lengths have lesser error than the estimation of precipitation amounts), as well as two sentences on the applicability of the developed methodology to future works (climate change, feed indicators to monitor drought impacts on hydrology and agriculture, etc.).

1. We added "meteorological drought" to the title.

[revised manuscript text omitted]